# In Silico Analysis of the Structural Dynamics and Substrate Recognition Determinants of the Human Mitochondrial Carnitine/Acylcarnitine SLC25A20 Transporter

**DOI:** 10.3390/ijms24043946

**Published:** 2023-02-15

**Authors:** Andrea Pasquadibisceglie, Virginia Quadrotta, Fabio Polticelli

**Affiliations:** 1Department of Sciences, University of Roma Tre, 00146 Rome, Italy; 2National Institute of Nuclear Physics, Roma Tre Section, 00146 Rome, Italy

**Keywords:** mitochondrial carriers, SLC25A20, carnitine/acylcarnitine carrier, structural dynamics, substrate recognition, AlphaFold 2, molecular dynamics, molecular docking

## Abstract

The Carnitine-Acylcarnitine Carrier is a member of the mitochondrial Solute Carrier Family 25 (SLC25), known as SLC25A20, involved in the electroneutral exchange of acylcarnitine and carnitine across the inner mitochondrial membrane. It acts as a master regulator of fatty acids β-oxidation and is known to be involved in neonatal pathologies and cancer. The transport mechanism, also known as “alternating access”, involves a conformational transition in which the binding site is accessible from one side of the membrane or the other. In this study, through a combination of state-of-the-art modelling techniques, molecular dynamics, and molecular docking, the structural dynamics of SLC25A20 and the early substrates recognition step have been analyzed. The results obtained demonstrated a significant asymmetry in the conformational changes leading to the transition from the c- to the m-state, confirming previous observations on other homologous transporters. Moreover, analysis of the MD simulations’ trajectories of the apo-protein in the two conformational states allowed for a better understanding of the role of SLC25A20 Asp231His and Ala281Val pathogenic mutations, which are at the basis of Carnitine-Acylcarnitine Translocase Deficiency. Finally, molecular docking coupled to molecular dynamics simulations lend support to the multi-step substrates recognition and translocation mechanism already hypothesized for the ADP/ATP carrier.

## 1. Introduction

The Carnitine-Acylcarnitine Carrier (CAC) is a member of the mitochondrial Solute Carrier Family 25 (SLC25), known as SLC25A20. Similar to other members of the SLC25 family, it is characterized by three homologous domains, each consisting of an even-numbered transmembrane ™ helix, an odd-numbered TM helix, and a third short helix, parallel to the matrix membrane layer, connecting the other two. Another feature of the family is the presence of the conserved motif signature PX[DE]XX[KR], located on the even-numbered TM helices and thus repeated in all of the three domains [1,2,3]. The charged residues of this motif form a salt bridge network on the matrix side of the carrier when the protein is open towards the intermembrane space (cytoplasmic state, c-state). A second, less conserved motif is [FY][DE]XX[KR], whose residues are involved in the formation of the cytoplasmic salt bridge network responsible for the stabilization of the transporter when the protein is open towards the matrix (matrix state, m-state) [4]. The SLC25 members are also characterized by the presence of a common substrate binding site consisting of conserved residues known as contact points (CPs), located on the three even-numbered TM helices [5].

CAC is involved in the electroneutral exchange of acylcarnitine (from C2 to C18) and carnitine across the mitochondrial inner membrane [6,7]. This mechanism of transport, also known as “alternating access”, involves a conformational transition in which the binding site is accessible from one side of the membrane or the other (c-state and m-state). Unlike other members of the family, CAC can promote the uniport of carnitine, even though this occurs at a lower rate compared to the antiport [6,7].

CAC is part of the carnitine shuttle, a fundamental cellular mechanism through which fatty acids are carried into the mitochondria for energy production via β-oxidation [6]. The carnitine palmitoyltransferase-1 (CPT I) is the first component of this shuttle and it is located in the outer mitochondrial membrane, where it catalyzes the conversion of long-chain acyl-CoA and carnitine into long-chain acylcarnitine and coenzyme A [8]. The trans-esterified acylcarnitines are transported from the cytosol to the intermembrane mitochondrial space (IMS) through the voltage-dependent anion channel (VDAC) [9] and then from the IMS to the matrix through CAC. Once in the matrix, the acyl groups of acylcarnitines are transferred back to coenzyme A by the last component of this shuttle, CPT II, which forms a supramolecular complex in the inner mitochondrial membrane with CAC [10]. The acyl-CoAs produced are then available for β-oxidation and the carnitine released returns into the intermembrane space through CAC and is available again for the transport of fatty acids [11].

Fatty acid β-oxidation (FAO) provides the main source of energy during prolonged starvation and for cardiac and skeletal muscles during long-term exercise. Since CAC is an essential component of the carnitine cycle, its deficiency or mutation of essential residues causes Carnitine-Acylcarnitine Translocase Deficiency (CACTD) in the neonatal period, with rapidly progressive deterioration and death in infancy or childhood [12]. CAC’s decreased protein-expression levels have been correlated with poor survival in patients with hepatocellular carcinoma (HCC). In particular, CAC down-regulation promotes HCC growth and metastasis development through the suppression of FAO. For this reason, it has been suggested that CAC could be used as a therapeutic target or as a prognostic factor for hepatocellular carcinoma [13]. Additionally, SLC25A20 rs7623023 polymorphism has been found to be connected to an increased risk of colorectal cancer insurgence associated with the consumption of red and processed meat [14].

Given the potential biomedical interest, the current work aims to characterize the dynamics of the SLC25A20 through an in silico approach. In this regard, MD simulations have been frequently used to investigate the molecular mechanism of the MCF members [15,16,17,18,19,20,21,22,23]. In this work, AlphaFold2 has been used to predict a structural model of the c-state of the transporter. Furthermore, a recently proposed AlphaFold2 implementation, which utilizes sub-sampling of the multiple sequence alignment, has been used to predict a reliable model of the m-state three-dimensional structure. These models were used in molecular dynamics (MD) and molecular docking simulations to determine key residues involved in the conformational change of the carrier and in early ligand recognition and binding. Indeed, several studies in recent years suggested that residues located at different depths of the carrier cavity establish interactions with the substrate, helping its migration deeper into the cavity [2,24,25,26]. An early ligand recognition mechanism has also been hypothesized for the ADP/ATP carrier on the basis of MD simulations [17,22], confirmed by experimental studies [27].

## 2. Results and Discussion

### 2.1. Analysis of the Cytoplasmic and Matrix State Structural Models

The new implementation of AlphaFold2, proposed by Del Alamo and colleagues [28], allowed for the obtainment of two main conformations of the human SLC25A20: the c-state, where the transporter is accessible from the IMS; and the m-state, where the transporter is accessible from the mitochondrial matrix (Appendix A).

As a member of the SLC25 family, SLC25A20 displays two sets of charged residues, on the cytoplasmic and matrix side of the protein respectively, that can form the cytosolic or matrix network in the m- and c-state, respectively, along the substrate translocation path. The residues known to be part of the cytoplasmic network are Gly94, Lys97, Glu191, Lys194, Glu288, and Met291, while the residues of the matrix network are Asp32, Lys35, Glu132, Lys135, Asp231, and Lys234 (Figure 1) [6].

Together with the cytoplasmic network, a belt of aromatic and hydrophobic residues (the hydrophobic plug) form the cytoplasmic gate involved in the stabilizing of the m-state conformation and in preventing the proton leak [4]. The aromatic belt residues have also been suggested to be involved in the correct folding and insertion in the bilayer membrane [29], whereas the presence of tyrosine residues could stabilize the cytoplasmic network by establishing hydrogen bonds with the negatively charged residues [4].

Since they are considerably conserved in the SLC25 family, the residues of this hydrophobic plug have been identified through a structural superimposition between SLC25A20 m-state and the crystallized structure of the ADP-ATP carrier from *Thermothelomyces thermophilus* (PDB ID: 6GCI) [4]. The corresponding residues on SLC25A20 resulted in the triplets Cys89-Phe90-Phe93, Tyr186-Phe187-Tyr190, and Cys283-Phe284-Phe287, located on even-numbered helices (Figure 1). Some of the residues proposed to be involved in substrate binding are located in the central region of the transporter, specifically Arg178, Asp179, Trp224, and Arg275 for carnitine [30], and Val25, Pro78, Val82, Met85, and Cys89 for acylcarnitines (Figure 1) [5,31].

The residues cited above are extremely superimposable on the orthologous residues of the ADP/ATP crystal structures (Appendix A), confirming the reliability of the produced SLC25A20 models.

### 2.2. Structural Asymmetry and Intraprotein Interactions

The two predicted structures were simulated with classical MD for a total of 2 μs each (1 μs for each independent replica, see Section 3 for details). The produced MD trajectories were then analyzed to study the geometry of the six TM helices (H1–H6) and to identify the critical interactions between the three domains.

#### 2.2.1. TM Helices Interfaces and Geometry: C- and M-State Comparison

A clustering procedure was performed to extract representative structures from the MD trajectories. In detail, the MD simulation frames were clustered based on the RMSD of the α-carbons of the TM helices (Appendix A). The centroid of the most populated cluster, both for the c- and m-state, was analyzed.

In Figure 2, the molecular contacts between the six TM helices, identified using Protein Contact Atlas, are schematically depicted. Most intraprotein contacts occur between flanking TM helices, and only a few are established among the even- or odd-numbered helices, which are mainly involved in the cytoplasmic or matrix network of salt bridges.

Analysis of the intraprotein interactions highlighted large differences between the two conformational states (Figure 2, Table 1). In particular, the three interfaces H1–H2, H5–H6, and H6-H1 are those that mainly differ in terms of the number of atomic contacts, suggesting a different packing between these TM helices involved in the conformational transition (Table 1). In detail, the number of atomic contacts between the helices H1–H2 and H5–H6 increases in the m-state conformation, while it decreases for the H6–H1 interface.

Considering the above, we performed a structural superimposition of the two states, selecting only the protein regions identified. This was performed using the *MatchMaker* tool implemented in the software Chimera v1.14 [33]. In detail, the superimposition between the two conformational states, corresponding to the H3-H4-H5 TM helices (Figure 3A,B), resulted in an RMSD of 2.78 Å. On the contrary, the superimposition of H6-H1-H2 yielded a higher RMSD of 4.64 Å. This result is further supported by the superimposition between the crystallographic structures of the c-state (PDB ID: 4C9H [34]) and m-state (PDB ID: 6GCI [4]) of the ADP/ATP carrier (Figure 3C,D). In detail, the superimposition of the H3-H4-H5 TM helices produced an RMSD of 4.59 Å, while the superimposition of H6–H1–H2 resulted in an RMSD of 6.56 Å.

These data indicate that the conformational transition mainly involves the TM α-helices H1, H2, and H6. This observation is coherent with the crystallographic structure of the m-state conformation, in which the first domain is displaced with respect to the other two, although it should be noted that this distortion could be caused by the binding of the inhibitor [4]. Moreover, NMR studies of the ADP/ATP transporter and GDP/GTP transporter suggested that the H6, H1, and H2 helices are more dynamic than the rest of the protein domains [35,36]. Finally, asymmetry in the MCs’ intraprotein interactions and in the protein-cardiolipins interactions have also been observed in several MD simulation studies [19,20,37,38,39,40].

Of note, the superimposition between the H6–H1–H2 α-helices also pointed out a large movement of h56, the short matrix helix linking H5 to H6, further supporting the larger movement of these three α-helices. In this regard, it should be noted that the mutation of the residue Gly268 located at the N-terminal of H6, corresponding to the loop linking H6 to h56, is responsible for the complete loss of function of SLC25A20 [41]. This residue, like other glycine residues, is located in the second part of the EGxxxxAr[KR]G motif and forms part of the PG-levels. Interestingly, a role for these glycine residues in the conformational changes of the transporter has already been predicted by Palmieri and Pierri [2], and supported by evolutionary comparative analyses [29] and in vitro studies [41]. Furthermore, in the m-state conformation, a close packing between Ala281, located on H6, and Leu213 and Gly217, located on H5, is observed. This provides a likely explanation for the pathogenic role of the Ala281Val mutation [42]. In fact, the presence at position 281 of a bigger hydrophobic residue likely perturbs the packing of the m-state helices, thus having a detrimental effect on the transition from the c- to the m-state.

To further understand the physical properties of the TM helices, the α-carbon RMSF, the bending angles, and the dihedral angles were analyzed during the MD trajectories (Appendix A). All of the six α-helices are characterized by a bending angle with an average value of 20 degrees. In detail, the odd-numbered helices display a kink between the 15th and 16th residues, while the even-numbered display a kink between the 5th and 6th residues. Notably, these positions are four residues (one α-helix turn) apart from the conserved proline residues, which are responsible for the bending of TM helices.

The biggest difference between the c- and m-state is in the bending angle of H2. In the c-state, H2 displays a strong kink with regards to residues Ile80 and Thr83, while in the m-state, H2 has only a slight curvature. A structural distortion of the H2 N-terminal half has also been observed by Brüschweiler and colleagues [35] in the case of the ADP/ATP transporter, supporting a critical role for H2 in the conformational transition.

The remaining TM helices conserved a similar bending. Minimal differences were observed at the C-terminal of H3 and H5, and at the N-terminal of H6; three regions located at the level of the matrix gate.

Interestingly, H3, H4, and H5 represent the most compact part of the protein structure in the m-state conformation, as also observed in the crystal structure of the ADP/ATP transporter from *Thermothelomyces thermophilus* [4].

#### 2.2.2. C-State Intraprotein Interactions

Although the cytoplasmic and matrix salt bridge networks represent the most stable interdomain interactions, different strengths and lifetimes are observed for each individual salt bridge.

The results of the c-state MD simulations demonstrated that Lys35 and Glu231 formed a salt bridge interaction, which is highly stable during MD1 (first replica) but oddly unstable during MD2 (second replica) (Figure 4, Table 2). In fact, the matrix network salt bridges experienced big fluctuations during MD2, suggesting that this replica sampled a disruption of the network (Figure 4D). Significantly, there is a change after 600 ns in the stability of the Asp32-Lys135 and Lys35-Asp231 salt bridges. This result indicates that the H3 TM helix interacts either with H1 or H5, but not simultaneously with both. Interestingly, experimental studies suggested that Lys35 is involved in the coupling of the two uniport reactions in opposite directions [43].

The H1 residue Asp32 established a strong and stable salt bridge with Arg275 in both trajectories (Figure 4B). This is often observed as one of the strongest electrostatic interactions occurring in mitochondrial carriers [19,20,37,44]. Surprisingly, Giangregorio and colleagues observed that only the Arg275Ala mutant totally abolished the transport activity, while Asp32Ala showed a significant residual activity [30]. Moreover, the Asp32Ala mutation displayed a Km value similar to that of the wild-type protein, but a strong reduction in the Vmax. On the other hand, the Lys135Ala mutant showed a transport rate similar to that of the wild-type, but an increase in the Km value. These data are in agreement with the instability of the Asp32-Lys135 interaction, observed both in the present simulations and in other computational studies (Figure 4C,D) [16,20]. Moreover, the fact that the Lys135Ala mutation is better tolerated than the Asp32Ala mutation could be linked to the Asp32-Arg275 interaction, as Arg275 is known to be crucial for the substrate translocation [30].

It is worthwhile to note that, in the c-state, some interactions between the odd-numbered helices are also present. These contacts involve residues of the hypothetical hydrophobic plug and an additional hydrogen bond between Tyr186 and Asn280.

Analysis of the MD simulations showed that, in the c-state, the hydrophobic contact between Phe86 and Cys283 is the only stable interaction, observed in both replicas, connecting two of the three protein domains (Appendix A). Additional hydrophobic interactions are Phe93 and Phe187, Phe90 and Phe187, and Phe86 and Phe287 (not shown).

#### 2.2.3. M-State Intraprotein Interactions

To date, very little data are available for the interactions of the cytoplasmic network residues. In this regard, the two MD simulations (MD1, first replica; MD2, second replica) of the transporter in the m-state conformation showed that the most stable cytoplasmic salt bridge is the one between Lys194 and Glu288, and that this interaction is stabilized by the hydrogen bond between Tyr190 and Glu288 (Figure 5 and Appendix A; Table 3). This is coherent with experimental data that demonstrate the involvement of Lys194 and Glu288 in the transporter activity [30,43].

The m-state conformation appears to be also stabilized by a hydrophobic plug already observed in the structure of the bongkrekic acid-inhibited mitochondrial ADP/ATP carrier (PDB ID: 6GCI [4]). In detail, hydrophobic contacts resembling π-stacking interactions are formed between the residues Phe90-Phe287, Tyr186-Phe284-Tyr190, Phe93-Phe187. Of note, Phe93-Phe187 was also observed during the c-state simulations (Appendix A). Interestingly, the mutation Phe284Gly causes detrimental effects in the activity of the carrier [45,46].

Finally, during both the c- and m-state simulations, the electrostatic interactions between the residues Asp32-Arg275 and Glu132-Lys234 are maintained, providing a structural explanation for their experimentally observed critical role in carrier activity (Figure 5) [30]. Moreover, these interactions contribute to stabilizing the I-III and II-III domain interfaces, particularly in the m-state where the protein is widely open toward the matrix.

#### 2.2.4. Interactions Involving Substrate Contact Point Residues

The hypothetical central binding site residues establish interactions present during both the c-state and the m-state simulations. Apart from the already mentioned Asp32-Arg275 interaction, the Asp179 residue stably interacts with Arg178 (hypothetical CP II residues), which in turn establishes electrostatic interactions with Glu132 and Asp231. In particular, the Arg178-Glu132 interaction has a longer persistence in the c-state, whereas the Arg178-Asp231 is the more persistent electrostatic interaction in the m-state (Appendix A).

Trp224 and Arg275 form a cation-π interaction during the c-state simulations, with the aromatic ring and the guanidinium group planes parallel to each other (Appendix A). It is interesting to note that this interaction is also observed in the MD simulations of SLC25A29 in the presence of the substrate [44]. Furthermore, in the case of the SLC25A20 rat orthologue, it has been demonstrated that Trp224 is crucial for carnitine binding and substrate-induced gate opening [47].

### 2.3. Substrates Early Recognition Step

In order to identify early substrate binding sites, representative conformations were extracted from the c- and m-state MD simulations, and a clustering procedure was performed based on the RMSD of the α-carbons of the m- and c-gate residues, respectively. The conformations with a wider opening in the transporter funnel were used for molecular docking and subsequent MD simulations with the two substrates. This choice was dictated by the results of preliminary MD simulations following ligand docking, in which using less open structures led to the exit of the ligand from the transporter vestibule. When docking PCAR to the c-state model, the search box area was centered around the residues of the cytoplasmic gate but encompassed the entirety of the external vestibule of the protein. Analogously, when docking CAR to the m-state model, the search box area was centered around the residues of the matrix gate but encompassed the entire internal vestibule of the protein. The central binding site was not included in the search area as the purpose of docking simulations was to investigate the existence of early recognition sites for the PCAR and CAR substrates. Given that SLC25A20 mediates the translocation of carnitine (CAR) from the matrix toward the IMS, this molecule has been chosen to investigate a potential early binding site located in the matrix side of the transporter. Conversely, SLC25A20 also mediates the import of acylcarnitines (C2-C18) from the IMS [48]. In that case, a representative acylcarnitine, propionylcarnitine (PCAR), was chosen to identify a potential early binding site located in the cytoplasmic side of the protein. A simple 3 carbon acylcarnitine was chosen in order to not excessively increase the degrees of freedom of the ligand simulated.

#### 2.3.1. Phe287 and Tyr190 Are Involved in Acylcarnitine Early Recognition

During the MD1 simulation of the c-state, PCAR trimethylammonium established a cation-π interaction with Phe287 for 73% of the simulation time, in addition to the carboxyl group forming a hydrogen bond with Tyr190 for about 40% of the simulation time (Figure 6; Table 4, Appendix A). The hydrophobic carbons of PCAR also established several hydrophobic interactions, of which Phe90, Tyr190, and Phe287 are the most significant (Appendix A, Appendix A).

Distance analysis revealed that the distance between Phe287 and PCAR remained stable during the simulation, at approximately 5 Å, while the two PCAR carboxyl oxygens interacted alternatively with the hydroxyl group of Tyr190 (Appendix A).

Similarly, during the MD2 simulation, PCAR trimethylammonium established a cation-π interaction with Phe287 for 30% of the simulation time, whereas the carboxyl group formed a hydrogen bond with Tyr190 with a persistence of 33% (Figure 6; Appendix A). The ligand also established a series of less frequent contacts with hydrophobic residues such as Phe86, Phe187, Tyr190, and Phe284 (Appendix A, Appendix A).

Taken together, these results suggest a common recognition step involving the cation-π and hydrogen bond interactions with Phe287 and Tyr190, respectively.

Interestingly, Phe90, located on H2, aligns with other aromatic residues that may play an important role in the correct folding of human SLC25A20 (Figure 7) [29]. In addition, Tyr190 precedes the cytoplasmic network residue located on H4, and is the Tyr brace involved in stabilizing the Lys194-Glu288 salt bridge (Figure 5). Both residues are also close to the residues orthologous to Asn96 and Arg197 of the homologue ADP/ATP carrier from *Thermothelomyces thermophilus*, which were proposed as binding residues in an early ligand recognition step (Figure 7) [22,27]. Moreover, De Lucas and collaborators demonstrated that Phe284 plays a fundamental role in the transport of acylcarnitines, both in vitro and in vivo, most likely by being directly involved in substrate binding [45,46].

In this regard, it has been shown that the Phe86Ala mutation decreased SLC25A20 residual activity compared to the wild-type [31], and comparative studies indicated Phe86 as one of the conserved residues potentially involved in acylcarnitine binding [49]. An interesting fact to point out is that Phe86, based on the sequence alignment with the mitochondrial basic amino acids transporter (SLC25A29) corresponds to Asn73, which has been proposed as the first contact point residue based on previous studies (Figure 7) [44,50].

Surprisingly, Phe86 and Asn73 are strongly conserved in all SLC25A20 and SLC25A29 orthologues, supporting the important role of these residues in substrate recognition (Figure 7).

#### 2.3.2. Trp224 and Asn280 Are Involved in Carnitine Early Recognition

At variance with the PCAR-SLC25A20 simulations of the c-state, CAR remained bound to the transporter in the m-state during only one simulation (MD1). For this reason, the first simulation was extended by an additional 100 ns, and a second MD simulation (MD2) was performed using the coordinates of a representative frame extracted from the previous simulation as input.

During MD1, CAR trimethylammonium established a cation-π interaction with Trp224 for more than 90% of the simulation time, making this interaction one of the longest-lasting overall. Additionally, in MD2, this was the most stable interaction observed between the ligand and the protein (Figure 8, Table 5). In addition, the CAR carboxyl group and Trp224 also established a persistent hydrogen bond (Appendix A). Experimental data from Giangregorio and colleagues suggest that Trp224 and Lys35 are involved in coupling the matrix gate opening to the substrate uptake, but not to the substrate efflux [47], considering that wild type and mutantTrp224Ala display similar Km values for substrate efflux.

The data from the simulations could be reconciled with the above cited experimental observations hypothesizing that a similar Km does not necessarily mean that Trp224 is not involved in substrate recognition in the m-state, but that its role could be taken up in the mutant by nearby residues that could establish interactions of different nature with the substrate other than cation-π interaction.

An alternative explanation is that Trp224 could be involved in an early, not rate limiting, step of the substrate translocation and thus its mutation does not significantly affect the Km of the overall process, in agreement with the emerging hypothesis of a multi-step translocation mechanism in mitochondrial transporters [22,27]. Interestingly, Trp224 belongs to one of the triplets of the substrate-binding area sites [29].

The carboxyl and the hydroxyl groups of the ligand also established hydrogen bonds with Trp224 and Asn280, respectively (Figure 8, Table 5). De Lucas and colleagues, working on *Aspergillus nidulans* strains, demonstrated that the Asn280Gly substitution almost completely inactivates the carrier while the Asn280Gln conservative mutation causes a transport reduction of about 70% [46]. Taken together, these results support a fundamental role for Trp224 and Asn280 in the CAR binding and translocation by SLC25A20 [47].

Notably, these Trp and Asn residues are conserved in all SLC25A20 and SLC25A29 orthologues, but both align with glycine residues of bovine and fungal ADP/ATP carriers (Figure 7). Substitutions at these two positions appear to characterize members of different MCs subfamilies (e.g., SLC25A20 and SLC25A29 with respect to SLC25A4) [51].

During both simulations, CAR also established a series of less frequent hydrophobic interactions with Leu24, Phe86, Trp224, and Ala227 (Appendix A; Appendix A).

## 3. Materials and Methods

### 3.1. Sequence Analysis

The canonical human SLC25A20 amino acid sequence (UniProt ID: O43772) has been used to generate a multiple sequence alignment (MSA) with its orthologues. Moreover, the human SLC25A29 (UniProt ID: Q8N8R3) and its orthologues were also included in the alignment, as well as the sequences of the bovine and fungal ADP/ATP carriers’ crystallized structures. In detail, the selected sequences were: CRC1 from *Saccharomyces cerevisiae* (UniProt ID: Q12289), DIF-1 from *Caenorhabditis elegans* (UniProt ID: Q27257), SLC25A20 from *Branchiostoma belcheri* (UniProt ID: A0A6P5AHZ8), SLC25A20 from *Danio rerio* (UniProt ID: Q6P5K6), SLC25A20 from *Xenopus tropicalis* (UniProt ID: Q6P628), SLC25A20 from *Gallus gallus* (UniProt ID: R4GLG2), SLC25A29 from *Caenorhabditis elegans* (UniProt ID: Q8I4M0), SLC25A29 from *Branchiostoma lanceolatum* (UniProt ID: A0A8J9ZWH2), SLC25A29 from *Danio rerio* (UniProt ID: F1QHQ1), SLC25A29 from *Xenopus laevis* (UniProt ID: A0A1L8FA24), SLC25A29 from *Gallus gallus* (UniProt ID: E1C3I5), SLC25A4 from *Bos taurus* (UniProt ID: P02722) and AAC from *Thermothelomyces thermophilus* (UniProt ID: G2QNH0).

The initial residue Met1 of the hSLC25A20 sequence has been removed based on experimental data that demonstrate that this residue is removed post-translationally [52]. However, the unprocessed sequence numeration has been used in the entire manuscript text.

The multiple sequence alignment has been obtained using MUSCLE [53], visualized using Jalview and colored with the “zappo” style of Jalview [54].

### 3.2. Protein Structure Prediction

The canonical human SLC25A20 amino acid sequence (UniProt ID: O43772) has been used to generate a structural model of the transporter in the c-state using the locally installed AlphaFold v2.2.0 version, with default settings [55]. In addition, a recent implementation of AlphaFold2, which allows different protein conformations to be sampled [28,56], has been exploited to obtain the structural model of the transporter in the m-state using the ColabFold notebook AlphaFold2_advanced (https://colab.research.google.com/github/sokrypton/ColabFold/blob/main/beta/AlphaFold2_advanced.ipynb, accessed on 1 December 2022) [57]. In detail, the settings used were: max_msa_clusters = 32, which determines the number of randomly chosen sequence clusters provided to the AlphaFold2 neural network; and max_extra_msa = 64, which determines the number of extra sequences used to compute additional summary statistics. It must be noted that optimal values of these parameters depend on the particular target protein [56]. Therefore, the minimum values available in the AlphaFold2_advanced notebook were used, which were effective in modelling different conformations of the carrier. The number of random seeds was set to 8 to expand the number of obtained models to 40 and, finally, the number of recycles was set to one and the minimization option deactivated.

The TMAlign algorithm [58] has been used to compare the structural models to the known crystallized structures, bovine SLC25A4 (bANT-1, PDB ID: 1OKC) [3] and fungal ANT-1 from Thermothelomyces thermophilus (PDB ID: 6GCI) [4], corresponding to the c- and m-state conformations of the SLC25 family. Therefore, the structures with the highest TM-scores are the ones that better represent the two conformational states of the protein.

The three-dimensional structure of the c-state obtained from the AlphaFold2 default algorithm displayed a similar TM-score but a higher pLDDT (local distance difference test) as compared to the one obtained from ColabFold. Therefore, the former one was chosen for subsequent analyses.

The obtained structural models displayed a pLDDT value of 87.88 for the c-state and 86.21 for the m-state, confirming their reliability.

The pLDDT score is a per-residue confidence score that ranges from 0 to 100, where regions with a score of higher than 90 are modelled with high confidence, regions with a score between 90 and 70 have a discrete quality and regions ranging from 50 to 70 are low confidence. In Appendix A, the SLC25A20 structural models have been colored according to the pLDDT score using UCSF ChimeraX 1.14 [59].

The first residue Met1 has been removed and an acetyl group has been added in agreement with the available experimental data [52].

The residues’ protonation state was predicted using PDB2PQR v3.5.2, with the default PARSE force field, and the implemented PROPKA [60,61].

### 3.3. Molecular Dynamics Simulation

SLC25A20 structural models have been inserted in a bilayer phospholipid membrane mimicking the inner mitochondrial membrane (IMM), using the web server CHARMM-GUI (http://www.charmm-gui.org, accessed on 1 December 2022) [62]. The membrane composition was based on the IMM model published by the CHARMM-GUI team available at the CHARMM-GUI Archive (https://charmm-gui.org/?doc=archive&lib=biomembrane, accessed on 1 December 2022) [63]. Different concentrations and lipid tail composition are used to better represent the inner and outer IMM leaflets. In this model membrane, phosphatidylcholine is the most represented phospholipid species, followed by phosphatidylethanolamine and cardiolipin, the latter being more abundant in the inner layer. Water molecules, from the TIP3P model, were added on both sides of the membrane, forming two layers each 22.5 Å thick. The total system charge was neutralized by adding NaCl ions, reaching a physiological concentration of 0.15 M. The CHARMM36m force field [64] and the AMBER22 package [65] were used to perform the MD simulations of the assembled systems (~100,000 atoms, see Appendix A for representative compositions), following the CHARMM-GUI protocol. First, an energy minimization procedure, involving 2500 steps of steepest descent and 2500 steps of conjugate gradient, was performed. Positional restraints were applied on the protein residues (10 kcal mol^−1^ Å^−2^) and on the membrane (2.5 kcal mol^−1^ Å^−2^). The resulting minimized systems have then been simulated using the canonical NVT ensemble, reaching a final temperature of 310.15 K. Thereafter, isothermal-isobaric NPT ensemble simulations have been performed to equilibrate the pressure to 1 bar. During the thermalization and equilibration phases, the positional restraints have been gradually reduced. The equilibrated system, without restraints, was properly simulated using the NPT ensemble for a total of 1 μs. Each system was simulated in replica. The Langevin thermostat has been used for both NVT and NPT ensembles, while the Monte Carlo barostat with a semiisotropic pressure scaling has been used for pressure control [66]. All of the MD simulations have been performed in a periodic boundary system. For the long range non-bonded interactions, the Particle Mesh Ewald method [67] and a 12 Å cut-off, with a force switching region at 10 Å, were used. The MD simulations were performed with a time step of 2 fs, apart from the apoSLC25A20 simulation for which the hydrogen mass repartitioning (HMR) method [68] was used with a 4 fs time step.

### 3.4. MD Simulation Analyses

All the analyses have been performed using the CPPTRAJ package [69]. For the hydrogen bonds and salt bridges analyses, given the high number of interactions observed, only the interactions with a persistence higher than 10% of the simulation time have been reported to limit the description to the most relevant ones. The salt bridges analysis has been performed through the *hbond* command, considering only the charged residues, imposing a cut-off distance of 4 Å and removing the angle cut-off. Hydrophobic contacts were retrieved using the *nativecontacts* command. The RMSD was calculated on the backbone atoms, while the RMSF was calculated on the α-carbons. For the cation-π interactions analysis, the distance between the geometric center of the aromatic ring and the trimethylammonium group of the ligand, and the angle formed between the cation and the aromatic ring plane, were monitored.

The geometry of the six TM helices was analyzed through the *multidihedral* command of the CPPTRAJ package using the HELANAL module of the MDAnalysis python library [70,71].

Finally, raw data have been parsed and plotted using pandas and matplotlib Python libraries [72,73].

### 3.5. Clustering Procedure

The *gromos* algorithm implemented in the GROMACS 2022 package [74] was used to extract representative conformations of the transporter from the MD trajectories. In detail, the initial part of the trajectories, in which the RMSD was not equilibrated yet, were excluded from subsequent analyses. The representative c-state and m-state structures, used for the intraprotein analyses, were obtained through a clustering procedure based on the RMSD of the α-carbon of the TM helices (selected residues: H1, 10–40; H2, 76–100; H3, 110–144; H4, 172–197; H5, 209–239; H6, 269–297), choosing as RMSD cut-off 0.1 Å for the c-state, and 0.15 Å for the m-state. The cut-off was chosen with a general rule of thumb, looking for a compromise between the number of clusters (around 10–20 in total) and the number of clusters containing a single frame.

The representative structures, used for the protein-substrates complex prediction, were extracted through the clustering of the c- and m-state trajectories, based on the RMSD of the residues located in the cytoplasmic half (selected residues: 2–17, 90–117, 186–220, 284–301) and in the matrix half of the transporter (selected residues: 18–39, 56–89, 118–143, 153–185, 221–239, 248–283), respectively. The RMSD cut-off values were chosen with the same criteria described above.

### 3.6. Molecular Docking

Molecular docking simulations have been carried out to identify potential early substrates binding sites. For both the c- and m-state, conformations with a wider opening of the transporter funnel were used for molecular docking and subsequent MD simulations with the two substrates. This choice was dictated by the results of preliminary MD simulations following ligand docking, in which using less open structures led to the exit of the ligand from the transporter vestibule.

PCAR was docked on the selected structure of the c-state, centering the search box area around the residues of the cytoplasmic gate and encompassing all of the cytoplasmic half of the protein.

In detail, the gridbox consisted of 142, 107, and 66 grid points along the x, y, and z axis, respectively. The gridbox spacing (i.e., the space between two adjacent grid points) was set to 0.375 Å and the xyz coordinates of the grid center were set to 2 (x), 0 (y), and 0 (z). The receptor has been converted to *pdbqt* format using the *prepare_receptor* command from the ADFR software suite v1.0 [75]. The ligands were retrieved from the ZINC database [76] in *mol2* format and converted to *pdbqt* using the Meeko script *mk_prepare_ligand.py* (https://github.com/forlilab/Meeko, accessed on 1 December 2022). The molecular docking was performed with AutoDock Vina 1.2.0 [77] using the AutoDock4 scoring function and setting the exhaustiveness parameter to 32.

The resulting best pose had a score of −3.7 kcal/mol. The predicted SLC25A20-PCAR complex was used for the MD simulations described in Section 3.7.

For the m-state, the first centroid was extracted and docked with CAR, centering the search box area around the residues of the matrix gate and encompassing all of the matrix half of the protein. The gridbox consisted of 40, 40, and 40 grid points along the x, y, and z axis, respectively. The gridbox spacing was set to 0.375 Å and the xyz coordinates of the grid center were set to 1.9 (x), −0.8 (y), and −0.9 (z).

The docking procedure was performed as described for the SLC25A20-PCAR complex. In this case, the best docking pose, simulated in complex with SLC25A20, turned out to be unstable during the dynamics trajectory. Only the fifth docking pose, with a score of −2.6 kcal/mol, reached a stable binding mode during the MD simulations.

It must be noted that the central binding site was not included in the search area as the purpose of these simulations was to investigate the existence of early recognition sites for the PCAR and CAR substrates.

All the docking results were visually inspected with Chimera [78].

### 3.7. MD Simulation of the Protein-Ligand Complexes

The previously obtained SLC25A20-substrate complexes were embedded in a bilayer lipid membrane and subjected to MD simulations, using the same procedures described in Section 3.6. Regarding the positional restraints, the ligands were treated as a protein residue, applying the same constant force during the MD phases. The MD production runs of the SLC25A20-substrate systems were performed for a total of 100 ns, only to check the reliability of the binding pose found. The only exception is represented by the SLC25A20-CAR complex, due to the difficulties in obtaining a stable complex. In this case, the first simulation was extended for a total of 200 ns, while a second simulation (of 100 ns) was performed starting from a frame extracted from the first 100 ns.

## 4. Conclusions

The generation of the structural models of the two SLC25A20 conformational states, coupled to extended MD simulations, allowed for insight into the molecular determinants of the protein dynamics of this transporter. In particular, the results of this study evidenced a significant asymmetry of the conformational changes leading to the transition from the c- to the m-state, with the H6–H1–H2 helices experiencing notable conformational changes as compared to the H3–H4–H5 helices. Comparative analysis of the c- and m-state experimental structures of the ADP/ATP carrier, obtained both by crystallography and NMR [4,35], as well as NMR studies on the GDP/GTP transporter [36], indicates that this is possibly a feature shared at least by some other members of the SLC25 protein family and possibly by all of them, as also hypothesized in other studies [20,37,38,39]. Analysis of the MD simulations’ trajectories also allowed for a better understanding of the role of SLC25A20 pathogenic mutations as the basis of CACTD [42]. In fact, the data suggest that the pathogenic mutation Asp231His perturbs the matrix salt bridge Asp231-Lys35 and/or the Asp231-Arg178 interactions, which are involved in the stability of the c- and m-state. A relevant role for the former salt bridge has also been highlighted in the MD study of the homologous ADP/ATP transporter by Dehez and colleagues [16]. Further, analysis of the two conformational states of the transporter provides a likely explanation of the role of the pathogenic mutation Ala281Val [42]. In fact, as Ala281 is located on the H6 TM helix at the interface with H5, the presence of a bigger hydrophobic residue could perturb the helices’ packing during the transition from c- to m-state, destabilizing the structure and causing detrimental effects on substrate transport.

Molecular docking and dynamics simulations in the presence of the substrates also allowed for the analysis of the early substrate’s recognition steps, identifying putative early recognition sites outside of the central binding site of the transporter (Figure 9). In fact, results obtained suggest that Asn280 and Phe284 interact with the ligands, the first establishing hydrogen bonds with CAR in the m-state and the second hydrophobic contacts with PCAR in the c-state. These data further corroborate the hypothesis of a multi-step transport process operating in the SLC25 family, already put forward for the ADP/ATP carrier [22,27] and other homologous transporters [2,24,25,26,29]. Finally, the information value of the approach employed in the present study is further confirmed by the role observed for Trp224, a residue known to be crucial for the antiport function of the transporter [47], and for whom an important role has also been predicted on the basis of MD simulations of SLC25A29 [44]. In fact, this residue has been predicted to establish cation-π interactions with both the PCAR trimethylammonium group, in the m-state, and the guanidinium group of the CPIII residue Arg275, in the c-state conformation.

## Figures and Tables

**Figure 1 ijms-24-03946-f001:**
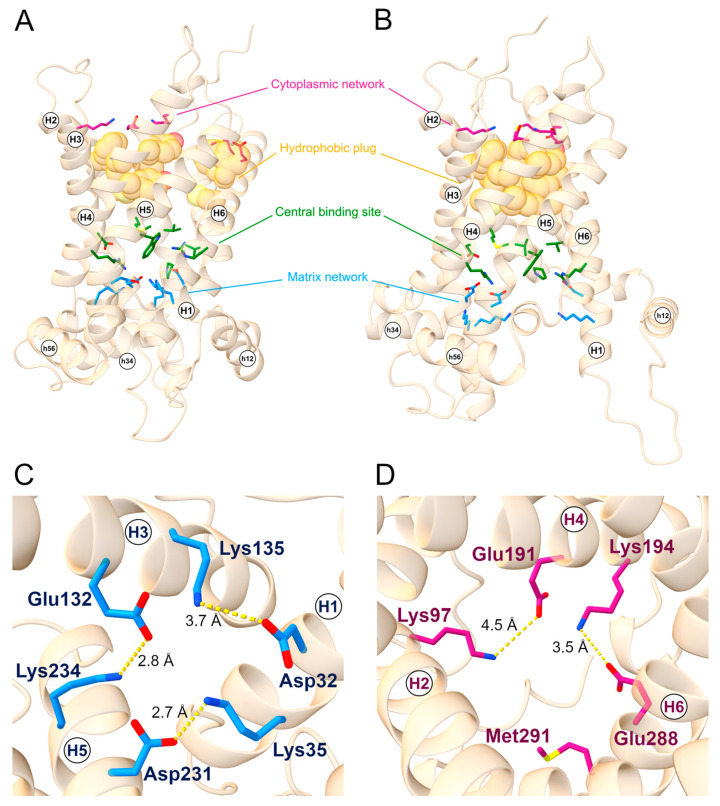
**SLC25A20 structural motifs.** General view of the (**A**) c-state and (**B**) m-state structural models of the SLC25A20 transporter. The cytoplasmic network residues are represented as sticks and colored in magenta; the hydrophobic plug residues are represented as van der Waals spheres and colored in orange; the proposed substrate binding residues are represented as sticks and colored in green; lastly, the matrix network residues are represented as sticks and colored in blue. (**C**) Top view of the c-state showing the m-gate residues together with ion-pairs distances. (**D**) Bottom view of the m-state showing the c-gate residues together with ion pairs distances. TM helices and short helices are labeled.

**Figure 2 ijms-24-03946-f002:**
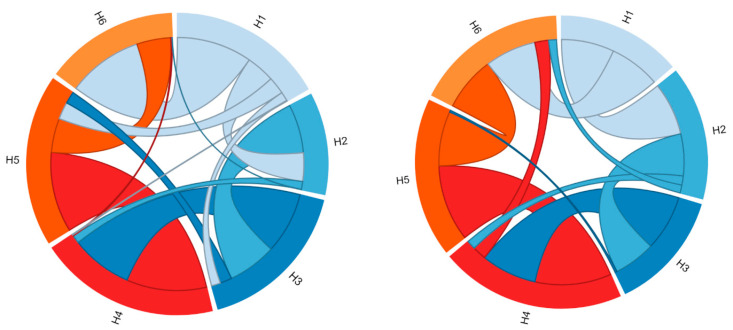
**Representation of the contacts between the six TM helices.** The two graphs represent the residue-residue interaction networks between the TM α-helices (H1–H6). On the left are represented the intraprotein interactions of the SLC25A20 c-state, while on the right those of the SLC25A20 m-state. The two graphs have been obtained with the Protein Contact Atlas tool [32].

**Figure 3 ijms-24-03946-f003:**
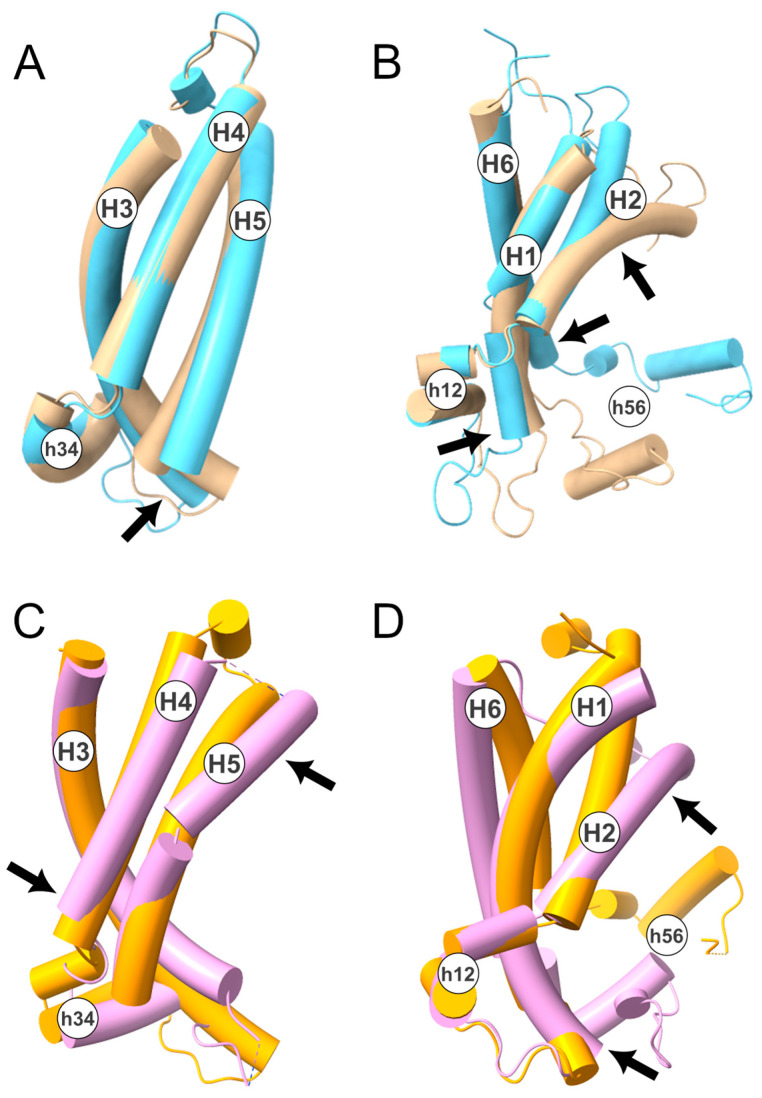
**TM helices superimposition between c- and m-state.** In the upper panels, is shown the superimposition of the TM helices H3–H4–H5 (**A**) and H6-H1-H2 (**B**) of the human SLC25A20 in c- (in beige) and m-state (in light blue). In the lower panels, is shown the superimposition between the TM helices H3-H4-H5 (**C**) and H6-H1-H2 (**D**) of the ADP/ATP carrier from yeast (PDB ID: 4C9H; c-state in orange) and from *Thermothelomyces thermophilus* (PDB ID: 6GCI [4]; m-state in pink). The four structures have been depicted with a tube representation and TM and short helices are labeled. Arrows indicate the regions that contribute the most to the calculated RMSD. In detail, for the SLC25A20 H3–H4–H5: 123–143 (RMSD 1–10 Å); H6–H1–H2: 35–39 (RMSD 2–4 Å); 80–99 (RMSD 1.5–16.5 Å); 268–277 (RMSD 1.5–8.5 Å); while for SLC25A4 the following (numbering of the yeast protein) H3–H4–H5: 192–196 (RMSD 25–32 Å); 228–240 (18.6–12 Å); H6–H1–H2: 58–60 (RMSD 22.5–26–5 Å); 102–116 (RMSD 8–26 Å).

**Figure 4 ijms-24-03946-f004:**
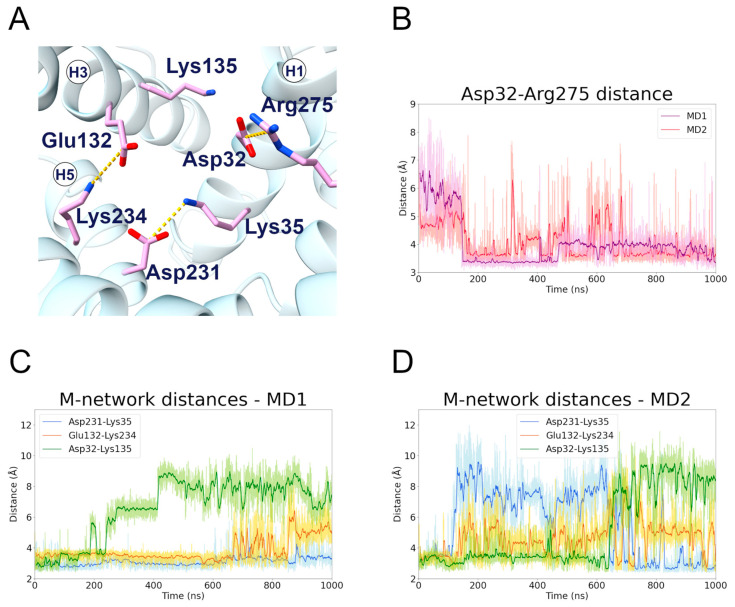
**Matrix network residues interactions along the c-state MD trajectories.** (**A**) Representative structure of the matrix network salt bridges formed during the c-state MD simulations; the interacting residues are represented in stick, colored in pink, and the salt bridge interactions are represented as yellow dashed lines. (**B**) Time series of the distance between the Asp32 and Arg275 side chains, during the two c-state MD simulations. (**C**,**D**) Time series of the distances between the side chains of the interacting pairs Asp32-Lys135, Glu132-Lys234, Asp231-Lys35, during the first (**C**) and second (**D**) MD simulation of the c-state.

**Figure 5 ijms-24-03946-f005:**
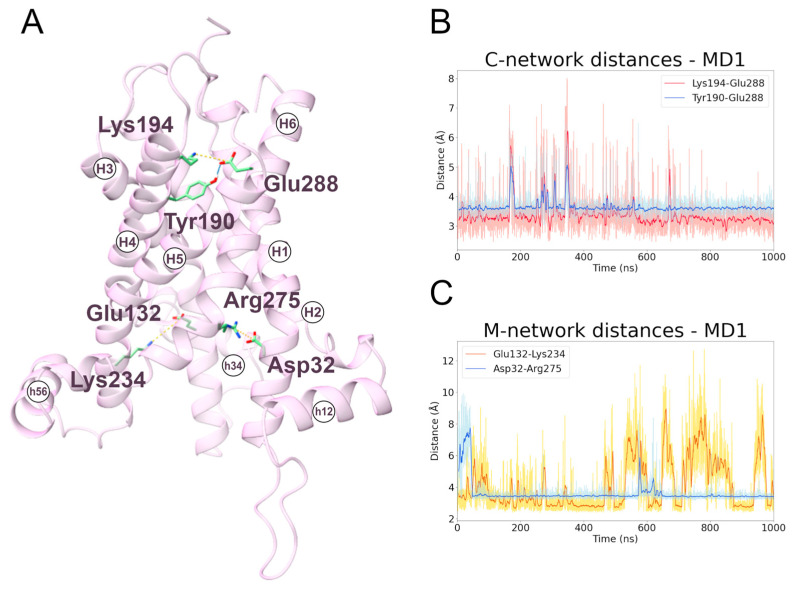
**Interactions between cytoplasmic and matrix network residues along the m-state MD1 trajectory.** (**A**) Representative snapshot extracted from the first MD simulation of the m-state. The SLC25A20 is represented as ribbons, in pink, while the interacting residues are shown as sticks, colored in green; salt bridges and hydrogen bonds are shown as yellow and cyan dashed lines, respectively. (**B**,**C**) Time series of the distances measured between c-network residues (**B**) and m-network residues (**C**) during the m-state MD1 trajectory.

**Figure 6 ijms-24-03946-f006:**
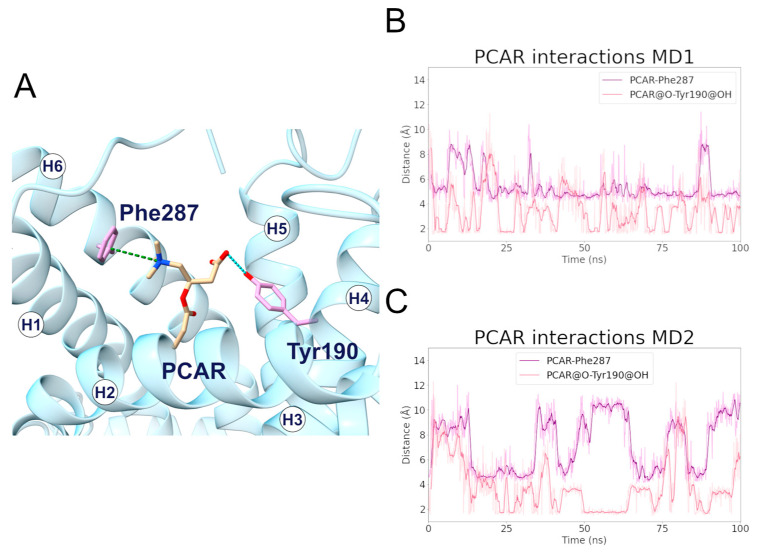
**Interactions established between SLC25A20 and the PCAR ligand.** (**A**) Representative snapshot of the SLC25A20-PCAR complex extracted from the first MD simulation. The SLC25A20 is represented as ribbons, in cyan, while the interacting residues are shown as sticks, colored in pink; the PCAR molecule is represented as sticks, colored in beige. (**B**,**C**) Time series of the distances measured between the PCAR atoms and the side chain of the interacting residues, during the first (**B**) and second (**C**) MD simulation. In the graph legend, the “@” symbol identifies the interacting atoms.

**Figure 7 ijms-24-03946-f007:**
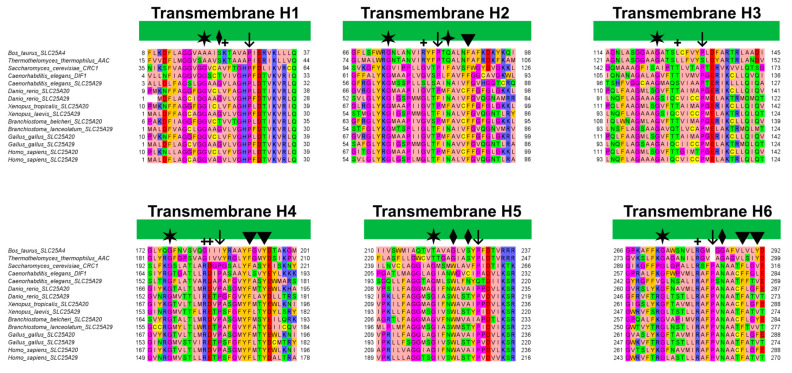
Multiple sequence alignment of the transmembrane helices of SLC25A20 and SLC25A29 orthologues with those of the bovine and fungal ADP/ATP carriers (PDB ID: 1OKC [3], 6GCI [4]). The alignment was obtained using MUSCLE and colored with the “zappo” style of Jalview. Diamonds and arrowheads indicate respectively the residues that align with the proposed early binding residues of CAR and PCAR; the four-point star indicates the alternating Asn and Phe residues. Plus symbols indicate putative contact points according to Giangregorio and colleagues; whereas arrows and stars indicate Pro and Gly residues proposed to be crucial for the tilting of the TM helices [41].

**Figure 8 ijms-24-03946-f008:**
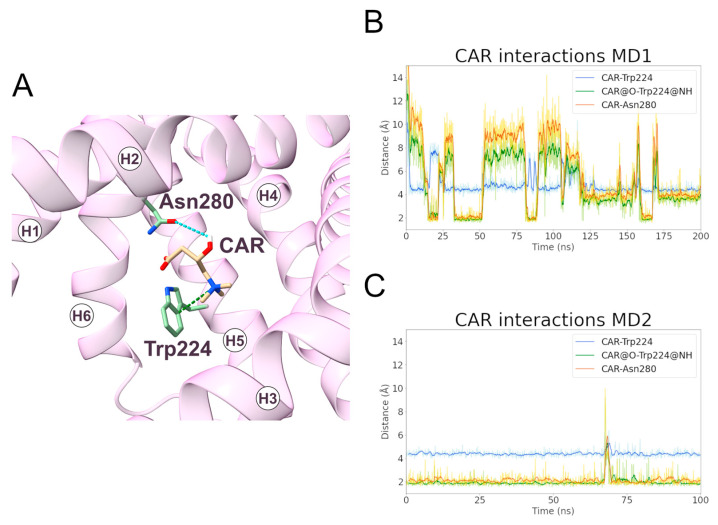
**Interactions established between SLC25A20 and the CAR ligand.** (**A**) Representative snapshot of the SLC25A20-CAR complex extracted from the first MD simulation. The SLC25A20 is represented as ribbons, in pink, while the interacting residues are shown as sticks, colored in green; the CAR molecule is represented as sticks, colored in beige. (**B**,**C**) Time series of the distances measured between the CAR atoms and the side chain of the interacting residues, during the first (**B**) and second (**C**) MD simulation. In the graph legend, the “@” symbol identifies the interacting atoms.

**Figure 9 ijms-24-03946-f009:**
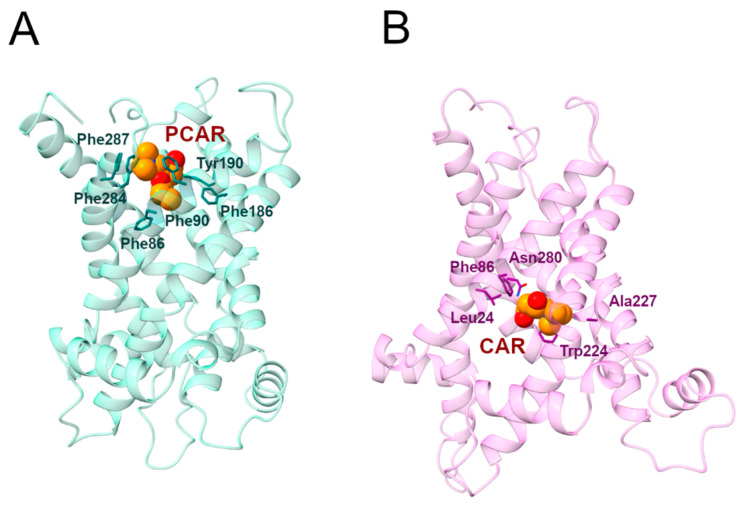
**Representative PCAR/CAR-SLC25A20 complexes.** (**A**) Representative snapshot extracted from the first MD simulation of the SLC25A20-PCAR complex. The SLC25A20 is represented as ribbons, in light blue, while the interacting residues are shown as sticks, colored in teal; PCAR is represented as van der Waals spheres with carbon atoms colored in orange. (**B**) Representative snapshot extracted from the first MD simulation of the SLC25A20-CAR complex. The SLC25A20 is represented as ribbons, in light pink, while the interacting residues are shown as sticks, colored in pink; CAR is represented as van der Waals spheres with carbon atoms colored in orange.

**Table 1 ijms-24-03946-t001:** Interhelices contacts of the SLC25A20 in the c-state and m-state, obtained with Protein Contact Atlas web-server.

	TM Helix	Residues Range	TM Helix	Residues Range	Total Atomic Contacts
**c-state**	H1	13–29	H2	78–88	269
	H2	77–100	H3	112–135	462
	H3	110–132	H4	172–195	582
	H4	174–197	H5	211–238	759
	H5	209–228	H6	269–285	323
	H6	270–295	H1	12–38	714
**m-state**	H1	10–29	H2	78–98	524
	H2	85–100	H3	112–128	415
	H3	110–132	H4	172–195	551
	H4	174–197	H5	211–238	756
	H5	209–228	H6	269–288	541
	H6	275–295	H1	12–32	505

**Table 2 ijms-24-03946-t002:** **Electrostatic interactions during the MD simulations of the SLC25A20 c-state.** For each salt bridge interaction shown in Figure 4, the fraction of frames in which it is present and the average distance are reported. For each pair of residues only the interacting atoms with the higher fraction have been included.

MD1			
Residue 1	Residue 2	Fraction	Avg Distance (Å)
Asp231	Lys35	0.94	2.9
Asp32	Arg275	0.84	3.0
Glu132	Lys234	0.50	2.8
Asp32	Lys135	0.20	2.8
**MD2**			
**Residue 1**	**Residue 2**	**Fraction**	**Avg Distance (Å)**
Asp32	Arg275	0.75	3.0
Asp32	Lys135	0.48	2.9
Asp231	Lys35	0.36	2.9
Glu132	Lys234	0.28	3.1

**Table 3 ijms-24-03946-t003:** **Electrostatic interactions during the MD simulations of the SLC25A20 m-state.** For each electrostatic interaction shown in Figure 5, the fraction of frames in which it is present, and the average distances are reported. For each pair of residues only the interacting atoms with the higher fraction have been included.

MD1			
Residue 1	Residue 2	Fraction	Avg Distance (Å)
Asp32	Arg275	0.94	3.0
Glu288	Lys194	0.78	2.9
Glu288	Tyr190	0.61	2.7
Glu132	Lys234	0.57	3.0
**MD2**			
**Residue 1**	**Residue 2**	**Fraction**	**Avg Distance (Å)**
Asp32	Arg275	0.84	3.0
Glu288	Lys194	0.70	3.0
Glu132	Lys234	0.51	3.0
Glu288	Tyr190	0.46	2.7

**Table 4 ijms-24-03946-t004:** **Electrostatic interactions during the MD simulations of the SLC25A20-PCAR complex.** For each electrostatic interaction shown in Figure 6, the fraction of frames in which it is present, and the average distances are reported. In the case of the cation-π and hydrogen bond interactions, the average angle is also reported. For each pair of residues, only the interacting atoms with the higher fraction have been included.

MD1				
Residue 1	Residue 2	Fraction	Avg Distance (Å)	Avg Angle (°)
Phe287	PCAR	0.73	4.8	88.7
PCAR	Tyr190	0.39	2.7	163.7
**MD2**				
**Residue 1**	**Residue 2**	**Fraction**	**Avg Distance (Å)**	**Avg Angle (°)**
Phe287	PCAR	0.31	4.9	85.4
PCAR	Tyr190	0.33	2.7	164.3

**Table 5 ijms-24-03946-t005:** **Electrostatic interactions during the MD simulations of the SLC25A20-CAR complex.** For each electrostatic interaction shown in Figure 8, the fraction of frames in which it is present, and the average distances are reported. In the case of the cation-π and hydrogen bond interactions, the average angle is also reported. For each pair of residues, only the interacting atoms with the higher fraction have been included.

MD1				
Residue 1	Residue 2	Fraction	Avg Distance (Å)	Avg Angle (°)
Trp224	CAR	0.79	4.6	93.2
CAR	Asn280	0.17	2.8	160.8
**MD2**				
**Residue 1**	**Residue 2**	**Fraction**	**Avg Distance (Å)**	**Avg Angle (°)**
Trp224	CAR	0.61	4.4	90.7
CAR	Asn280	0.51	2.8	160.0

## Data Availability

Data are available at https://doi.org/10.5281/zenodo.7467357 (accessed on 1 February 2023).

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
