# Peer review of "In Silico Analysis of the Structural Dynamics and Substrate Recognition Determinants of the Human Mitochondrial Carnitine/Acylcarnitine SLC25A20 Transporter"

_ijms, 2023, doi:10.3390/ijms24043946_

Round 1

Reviewer 1 Report

Major concerns:

1. 1.     The carnitine/acylcarnitine carrier (CAC) transports carnitine and acylcarnitines with acyl chains from 2 to 18 carbon atoms. The mammalian transporter exhibits higher affinity for acylcarnitines with longer carbon chains (Indiveri 2011). As ligand in virtual docking and molecular dynamics, authors used carnitine and propionylcarnitine, respectively with the m-state and c-state of CAC. Authors should provide an explanation for the use of these molecules as ligands, especially for propionylcarnitine.

2.  Ala281Val mutation is discussed only in conclusions at lane 511 without any supporting results.

Minor points:

1. The abbreviation CP, for Contact Points, is used at lanes 252 and 526, but it was not defined at first mention. It would also be appropriate to report the definition of contact point for this carrier family (Kunji 2006 or more recent);

2.  Legend to figure 1, A and B description are missing.

3. Legend to figure 5, B and C description are missing;

Reviewer 3 Report

The authors present a computational analysis for investigating conformational changes and substrate translocation in the carnitine/acyl-carnitine carrier. Although the presented results may sound, the authors need to discuss their findings in the context of important literature data that were missed during manuscript preparation. Some concerns arise about the preparation of CAC 3D models by using AlphaFold, without checking the correct structural alignment of the generated 3D models with the available crystallized structures for verifying the occurred alignment of the well-characterized structural features of mitochondrial carrier family members. Other concerns arise about docking analysis whose methods need to be described in greater detail before this analysis can be evaluated. The authors have the ability to address all the reported concerns, as long as all the presented analyses will be re-evaluated in light of the below reported minor and major concerns and the related literature.

INTRODUCTION

Line 36: In place or in addition to Reference [1] cited at line 36, reporting about human mitochondrial carriers, this reviewer would consider appropriate citing a more general/appropriate review about mitochondrial carrier family features highlighted by comparing mitochondrial carriers from several species. In this regard you can read:

Palmieri and Pierri, FEBS 2010: https://pubmed.ncbi.nlm.nih.gov/19861126/

In addition, the authors should also anticipate at this level the citation of the paper about the first mitochondrial carrier structure (the one with the highest resolution), which allowed to shed light on the structural features of mitochondrial carrier repeats for the first time:

Pebay-Peyroula et al., NATURE 2003: https://pubmed.ncbi.nlm.nih.gov/14603310/

Line 37 and line 40: “salt bridges network” should be “salt bridge network”;

Line 54: a reference about VDAC should be provided. This reviewer would suggest:

De Pinto, BIOMOLECULES 2021: https://pubmed.ncbi.nlm.nih.gov/33467485/

RESULTS and DISCUSSION

Section 2.1

Line 89: the sentence “SLC25A20 is characterized by two salt-bridge networks, on the cytoplasmic and matrix side of the protein, respectively” is not correct. Indeed, mitochondrial carriers show two sets of charged residues that can form the matrix gate or the cytosolic gate in two different conformations along the substrate translocation or transport cycle. The two gates are not formed simultaneously, as the above-cited sentence appears to indicate.

Line 93:

In the introduction section, the authors speak about “reliable models”. While it is clear why the authors chose to use AlphaFold 3D models for their analysis, according to the known limits of alphafold (see i.e., https://www.embopress.org/doi/full/10.15252/embr.202154046) this reviewer believes that a more robust comparative analysis with the available crystallized structures should be performed before using the 3D models for MD analyses. In this context, this reviewer would suggest providing a “structural” alignment of the obtained 3D models in the c-conformation and in the m-conformation and the corresponding portions in the crystallized structures to verify the structural alignment of

1. residues of the sequence motif PX[DE]XX[KR]X[KR]……EGXXXXAr[KR]G,

2. residues of the motif F[DE]XX[KR];

3. residues of the contact points of mitochondrial carrier binding similarly located substrate binding site.

The structural alignment of the cited portions should be added in fig. 1 or in a dedicated supp. Fig.

In addition, the authors should calculate the RMSD of the backbones of the generated 3D models with the backbones of 1okc (for the c-conformation) and 6gci (for the m-conformation). Furthermore, this reviewer would suggest checking that the 3D models show charged residues of the matrix gate in the c-conformation and of the cytosolic gate in the m-conformation are in the correct distance-range to form the right inter-repeat ion pairs.

In this regard, please, add to Figure 1 two further panels showing a top (zoomed) view of the c-state 3D model showing the m-gate residues (that should be labeled) with ion-pair distances and a bottom (zoomed) view of the m-state 3D model showing the c-gate residues (that should be labeled) with ion pair distances. Please, integrate the related Figure legend accordingly.

Both, methods and results of these estimations should be introduced in the corresponding sections. In this regard, the authors should read, cite and take into consideration for their discussion the following paper:

Pierri, Palmieri, De Grassi, Cell Mol Life Sci 2014: https://pubmed.ncbi.nlm.nih.gov/23800987/ (and the related supporting information);

It is not clear why the models reported in Figure 1 look like different from the models reported in Figure S1. Does it depend on the different orientation of the models in the 3D Mol Visualizer? Od did the authors use different models for the two figures? The authors should number transmembrane helices and short helices parallel to the membrane plane in Figure 1 and Supp. Fig. 1 according to the above reported Pierri et al, CMLS2014.

Line 94-96: it is not clear what the authors mean when they state

“Below the cytoplasmic network, several hydrophobic residues form a barrier which prevents the access to the carrier cavity from the intermembrane mitochondrial space, in the m-state. “

First of all “intermembrane mitochondrial space” can be replaced with “intermembrane space”, often abbreviated with “IMS”;

Secondly, at line 95 the authors report “the access”.. the “access” of what? Please, integrate.

Lines 96-100: Furthermore, the m-state is the state open towards the matrix and closed towards the IMS. If the cytosolic gate is closed (as it is in the m-state), ligands cannot enter the mitochondrial carrier substrate translocation pathway from the IMS. Thus, in the above-reported sentence, it is not clear the function of the hydrophobic barrier residues claimed by the authors and how it may prevent substrate entry when the carrier is open towards the IMS or how this region can allow/prevent substrate entry along conformational changes.

Did the authors refer to the fact that mitochondrial carrier substrates are anions, which can be slowed down by the presence of hydrophobic residues? In this reviewer opinion, that would be a misinterpretation, difficult to be supported/demonstrated.

In this reviewer opinion, the region enriched in “aromatic” more than “hydrophobic” residues (below the c-gate residues and at the level of the m-gate residues) can play a different role. Most likely, in this regard, the authors should read, cite and take into consideration for their analysis and interpretation the below-reported paper

Pierri, Palmieri, De Grassi, Cell Mol Life Sci 2014: https://pubmed.ncbi.nlm.nih.gov/23800987/ (and the related supporting information)

that introduces the role played by residues between cytosolic gate residues and binding site residues in the formation of aromatic belts and in substrate recognition.

More in general this reviewer believes that the above-reported sentence needs to be reworded.

Section 2.2.1

Lines 120-150: it is not easy to follow the interactions claimed by the authors from the reported sentences and from Figure 2, when the authors report, i.e.,

“In the c-state, the N-terminal of the odd-numbered helices is in contact with the N-terminal of the preceding even-numbered helix, while the C-terminal contacts the C-terminal of the following even-numbered helix”..

It would be better to provide a table listing residues with the observed interactions both for the c-state and for the m-state.

It is not clear what the authors did when they state

“The superimposition between the two conformational states, in correspondence of the H3-H4-H5 transmembrane helices (Fig. 3A,B), showed no significant differences, with an RMSD of 2.78 Å”.

Do the authors mean that “2.78 Å” is not significant?? How did they obtain that RMSD? Did they cut the single helices before superimposing them? How did they superimpose the proteins and/or the cut portions?

Is the author's analysis based only on TM align?? Do they have any reference comparative analysis for stating that “2.78 Å” of RMSD is not significant compared to “4.64 Å”? TMalign highest score structures can be corrected from a chemical/physical point of view, but ineffective for the following analysis, if the authors do not provide the above-requested structural alignments (see comments above about sentences reported at line 93).

In this regard, what the authors mean when they report that

“This result is further supported by the superimposition between the crystallographic structures of the c-state (PDB ID: 4C9H [16]) and m-state (PDB ID: 6GCI [2]) of the ADP/ATP carrier (Fig. 3C,D).”

In this case, the authors report that they obtained an RMSD range between 4.59 and 6.56 Å.

The obtained RMSD can depend on the tools and commands they used to align/superimpose the compared conformations. Structural alignments/superimposition can depend on sequence amino acid composition or on the fold recognition algorithms. Did they try to produce RMSD by using the “align” or “super” tools implemented in PyMOL?

The observed great RMSD difference can depend on the alignment/superimposition tools or on the packing of the 3D models generated by alphafold. In both cases, it needs to understand which is the cause of the reported great RMSD and which are the different protein regions responsible for the observed RMSD, which should be highlighted with arrows et sim in Figure 3. It would be better to use different tools for the structural alignment to verify that the obtained results do not vary too much depending on the employed tools.

Figure 3:

Transmembrane helices and short helices parallel to the membrane plane should be labeled;

Figure 4A:

Transmembrane helices should be labeled;

Figure 5A:

Transmembrane helices and short helices parallel to the membrane plane should be labeled;

Figure 6A:

Transmembrane helices should be labeled;

Line 162:

The authors report that

“This observation is coherent with the m-state conformation of the ADP/ATP transporters [2], in which the first domain is displaced with respect to the other two.”

However, this reviewer retains that the published m-conformation asymmetric structure is due to the presence of the nanobody and it is not completely trustable above all the bottom half (including short helices). The authors should take into consideration this in the interpretation of their analysis.

Line 166:

The authors should indicate that G268 is also the second glycine of the second part of mitochondrial carrier sequence motif EGxxxxAr[KR]G and part of the PG-levels, whose role in conformational changes was predicted in Palmieri and Pierri FEBS2010 https://pubmed.ncbi.nlm.nih.gov/19861126/, then supported by a technical structural/evolutionary comparative analyses in Pierri et al., Cell Mol Life Sci 2014 (https://pubmed.ncbi.nlm.nih.gov/23800987/) and in vitro investigated by Giangregorio et al., IJBIOMAC 2022 https://pubmed.ncbi.nlm.nih.gov/36122779/ (the latter correctly cited at this level). In Pierri et al. CMLS2014 it was proposed the cis/trans isomerization of glycine residues of PG-levels is responsible for the conformational changes triggering the formation of the c-conformation or of the m-conformation.

Most likely the authors should discuss the role of G268 and other glycine residues of PG-levels as hinge points for conformational changes and substrate translocation, according to what they observed in their trajectories, also for the following discussion.

Sections 2.2.2 and 2.2.3

In the interpretation of MD analysis, the authors missed reading and discussing, in the context of their results, several important papers about MD analysis on the ADP/ATP carrier reported here below, whose results deserve to be used as a reference system. It is strongly recommended that the authors read those papers and take them into consideration for their discussion:

Pietropaolo et al., BBA2016: https://pubmed.ncbi.nlm.nih.gov/26874054/

Wang and Tajkhorshid, PNAS2008: https://www.pnas.org/doi/abs/10.1073/pnas.0801786105

Dehez et al., JACS2008: https://pubs.acs.org/doi/10.1021/ja8033087

Johnston et al., MMB2009: https://www.tandfonline.com/doi/full/10.1080/09687680802459271

Skulj et al., IJC 2020: https://onlinelibrary.wiley.com/doi/abs/10.1002/ijch.202000011

Yi et al., IJMS 2022: https://pubmed.ncbi.nlm.nih.gov/36142790/

Yi et al., Mitochondrion 2019: https://pubmed.ncbi.nlm.nih.gov/31129042/

Yi et al., JLR 2022: https://pubmed.ncbi.nlm.nih.gov/35569528/

Falconi et al., Proteins 2006: https://pubmed.ncbi.nlm.nih.gov/16988954/

Tamura and Hayashi, Plos One 2017: https://journals.plos.org/plosone/article?id=10.1371/journal.pone.0181489

Line 190 and 191: due to the presence of the methods section at the end of the manuscript the authors should briefly cite/explain in section 2.2.2 and 2.2.3 which are the differences between MD1 and MD2.

Labels and titles of Figure 4 and Figure 5 panels show some cut portions and shadows, maybe due to poor resolution. Maybe it depends on the automatic webserver conversion. Please, check.

Section 2.2.4

Lines 258-260: When the authors report

“Thus, the interaction between these two residues could affect their side-chain orientation and determine how they interact with the substrates as observed in the MD simulations of SLC25A29 and suggested also for the SLC25A20 rat orthologue [20,23]”

They should consider that the same presence of the substrate in the carrier cavity would influence the orientation of side chains protruding towards the carrier cavity, thus, the observed interactions in the presence and absence of the substrate should be discussed separately and not linked/compared as above-reported.

Section 2.3 and subsections

It is not clear how the authors chose the representative conformations extracted from the c- and m-state MD simulations. Based on what the authors can say that conformations with a wider opening of the transporter funnel (do they mean cavity?) are better conformations to be considered as starting conformations for docking analysis.

When the author state that propionyl-carnitine was involved in cation-π interactions with F287, do they mean that the “trimethyl-ammonium group” of propionyl-carnitine was always (80% of simulation) in contact with F287? Which is the length of π interactions observed along the simulations? Generally, it is assumed that π interactions are 2.5 angstroms long.

How much distant is Y190 from F287? In table S1-S2-S3 only hydrogen bonds, salt bridges, and hydrophobic interactions, respectively, are reported.

More in general, it would be appreciated that the authors report which functional groups of the docked ligands were involved in the claimed binding interactions with the cited residues, both in sections 2.3.1 and 2.3.2.

Why did the author choose propionyl-carnitine for docking with the c-state? Was their choice based on the known Km or IC50 estimations that make it the best CAC substrate? If yes, they should cite appropriate references.

In this reviewer opinion, the authors should check the position of CAC F90 and Y190 in their alignments and in their structural models. Indeed, considering that the second part of the sequence motif EGxxxxAr[KR]G is easy to be aligned among mitochondrial carriers, the authors can easily verify that CAC_G74 residue aligns with 6gci_G81 and 1okc_ G72 residues. Starting from this observation, the authors can easily verify that CAC_F90 aligns with 1okc_F88 (see also Giangregorio et al., IJBioMac2022) and with 6gci_F97. Similarly, they can easily verify that CAC_Y190 aligns with 1okc_Y194 and with 6gci_Y204.

Thus, more than “in positions close to those occupied by N96 and R197”, they should focus on the fact that Y190 is the aromatic residue preceding the residues of the c-gate on helix 4, whereas F90 aligns with other aromatic residues, which can play an important role in the correct folding of CAC (and other mitochondrial carriers), beyond the fact that some aromatic residues at that level can interact in the ADP/ATP carrier with the aromatic moiety of the translocated nucleotides. Please, check and discuss this point or reword the related sentence.

Can the authors show a multiple sequence alignment of the aligned portions of the modelled CAC and the sequences of the crystallized structures of the ADP/ATP carrier in c-conformation and in m-conformation?

In the multiple sequence alignment, the authors should also show SLC25A29 sequence to highlight the alignment of N73 with F86. It is suggested that the authors produce an inter-helix multiple sequence alignment as the ones shown by Giangregorio et al., IJBioMac 2022. This figure should become a figure of the main text in place of the Fig. S13.

Lines 323-325:

It is not clear what the authors mean when they state “Trp224 is not involved in substrate recognition in the m-state, but that its role could be taken up in the mutant by nearby aromatic residues, likely candidates being Y186 and F86”. Indeed, the m-state is open towards the mitochondrial matrix, and W224 is located at the level of residues of the mitochondrial carrier similarly located common substrate binding site, whereas F86 and Y186 are located between residues of the similarly located substrate binding site and residues forming the c-gate. Can you better argument the above reported sentence?

More in general, concerning substrate recognition and substrate specificity the authors should read, cite and take into consideration for their analysis and interpretation of their results the below reported paper

Pierri, Palmieri, De Grassi, Cell Mol Life Sci 2014: https://pubmed.ncbi.nlm.nih.gov/23800987/ (and the related supporting information)

that introduces the role played by several residues on odd and even transmembrane helices, as well as by residues of matrix loops, in substrate recognition and translocation for all mitochondrial carriers.

In addition, concerning the characterizing residues of the mitochondrial carrier subfamilies the authors should read, cite and consider for their discussion the above reported FEBS2010 and the below reported reviews:

Palmieri et al., TPJ 2011: https://pubmed.ncbi.nlm.nih.gov/21443630/

Palmieri and Pierri, Essays in Biochemistry 2010: https://pubmed.ncbi.nlm.nih.gov/20533899/

and the already cited papers from Kunji and Robinson (PNAS2006 and PNAS2008) and the related supp mat.

Line 328:

When the authors report on the multi-step translocation mechanism in mitochondrial transporters, they should know that it is not an emerging hypothesis, due to the fact that several papers of the past 15 (and more) years explained that residues of responsible for substrate specificity are localized in-between the c-gate and the m-gate. Those residues play a role in the recruitment, binding, and specificity of the transported substrates, While large substrates (as dinucleotides) can interact with residues on different depths within the carrier cavity, in the case of smaller substrates, it is likely that successive interactions occur with some residues of residues located at various levels of the MC cavity, help the descent of the substrate deeper into the cavity driven by a positive electrostatic potential existing at the bottom of the carrier cavity

This hypothesis is clearly reported in

Pierri, Palmieri, De Grassi, Cell Mol Life Sci 2014: https://pubmed.ncbi.nlm.nih.gov/23800987/

It was also reported that an early event in substrate translocation through the carrier protein, a substrate recognition event may occur (https://pubmed.ncbi.nlm.nih.gov/19861126/) as proposed for the ADP/ATP carrier whose cavity is lined by a tyrosine ladder that can be important for sliding the ADP to the binding site (it can be seen in the beautiful movies of the supp. Mat. published within https://www.pnas.org/doi/abs/10.1073/pnas.0801786105).

But the hypothesis of a multi-step translocation mechanism is also supported by papers reporting about a regulatory site, far from the contact points of the substrate binding site, that can modify substrate specificity or the kinetics of substrate translocation:

Park et al., Cancer Cell Int 2012: https://pubmed.ncbi.nlm.nih.gov/22448968/

Seccia et al., JoF2022: https://pubmed.ncbi.nlm.nih.gov/36012783/

Vozza et al., BBA2017: https://pubmed.ncbi.nlm.nih.gov/27836698/

Miniero et al. JCM2022: https://pubmed.ncbi.nlm.nih.gov/36556135/

This hypothesis is also reported in papers describing the behavior of missense mutations of PG-level residues and other residues located along the MC cavity able to affect substrate translocation (although in the case of amino acid replacement with similar amino acids) or alter mitochondrial carrier kinetics parameters in orthologous carriers from phylogenetically related organisms and in mitochondrial diseases. A small group of papers on the topic is reported below

Lunetti et al., BBA2013: https://pubmed.ncbi.nlm.nih.gov/23850633/

Ma et al., JBC2007: https://pubmed.ncbi.nlm.nih.gov/17400551/

Muller et al., Biochemistry 1997: https://pubmed.ncbi.nlm.nih.gov/9398336/

Wohlrab Biochemistry 2002: https://pubmed.ncbi.nlm.nih.gov/11863464/

This reviewer believes that the authors should read and discuss their results in light of the above-reported papers (and maybe other papers on the topic that cannot be cited in this referee report) that should also be mentioned in their discussion.

Line 334:

The critical role of Trp24 in substrate binding and translocation was already proposed and the dedicated references should be cited at this level.

METHODS

Section 3.1

Line 354: Ort-1 from S. cerevisiae is not ortholog from SLC25A20 or from SLC25A29. If you put it in the alignment, you should also consider mammalian ornithine carriers in your alignment. Please, remove it from the alignment of Figure S12, or explain that you used it as the “putative” closest paralog of SLC25A29 in S. cerevisiae.

More in general, SLC25A20, SLC25A29 and Ort1p show several differences in terms of substrate specificity and amino acid composition. It should also be considered that a similar (or the same) protein function does not imply orthology. Indeed, it is assumed that two orthologous sequences originate from a speciation event, and in absence of a dedicated analysis, it would be better to speak about the “closest homologs”.

Section 3.2

Line 365: When the author state that “The canonical human SLC25A20 amino acid sequence has been used to generate a structural model of the transporter in the c-state using the AlphaFold2” do they mean that they downloaded the available 3D model present on the alphafold database (https://alphafold.ebi.ac.uk/)? If yes, they should reword their sentence accordingly and they should also introduce the weblink in the section. Alternatively, they should provide a more detailed protocol with parameters used for generating their c-state CAC 3D model.

Section 3.3

Lines 404-415: In the method section about MD simulations, the authors should provide the exact size of the simulation box and the total number of atoms, and the number of each molecule of the simulation box (water, lipids, amino acid, cations, anions) and the MD parameters chosen for their analyses (equilibration and production) in order to allow readers to reproduce the exact conditions of their simulations. The starting coordinates of their system should be provided as a supplementary txt file.

Section 3.4

Line 442: with the ammonium group, do the author mean trimethylammonium? Please, check

Section 3.5

Line 453-458: can the authors provide more details about the excluded “initial part of the trajectories showing a not equilibrated RMSD” in this section or in the corresponding RESULTS section?

A low RMSD is not a trustable criterion for the choice of the representative c-state and m-state conformation of 3D-modelled homologous proteins, based on the backbone alignment. This author would suggest to perform a structural comparative analysis as above-requested (see the comments at line 93).

The authors report a group of selected residues for the aligned transmembrane helices (H1, 10-40; H2, 76-100; H3, 110-144; H4, 172-197; H5, 209-239; H6, 269-297). It is noticed that from the selected residues the authors compared helices of different lengths. They should argument this choice and cite a reference from which they take those numbers. Or, more correctly, they should refer to papers about crystallized structures or to some of the above-cited papers in order to select a more representative set of residues for the comparison of the transmembrane helices.

Line 462-464: it is not clear what the authors mean with “c-gate residues (selected residues: 2-17, 90-117, 186-220, 284-301) and m-gate residues (selected residues: 18-39, 56-89, 118-143, 153-185, 221-239, 248-283)”. Please, explain.

Section 3.6

The authors should explain why they chose “the centroid with a slightly more open conformation towards the intermembrane space” as the c-state to be used for docking analysis.

In addition, it is not clear what the authors mean when they state that the box to be explored was prepared by “setting the search box area around the residues of the cytoplasmic gate (in the c-conformation) and around te matrix gate (for the m-conformation)”. Why they focused on the c-gate or the m-gate and did not use a box extended from the c-gate to the m-gate or a box extended from the c-gate to the contact points. It is evident that based on a box to be explored you will determine a binding pose that is the best pose in the investigated box.

In this regard, which was the scope of their docking analysis? How did they validate the proposed box and docking strategy?

Alternatively, they should reword their sentences and explain what they did in a more detailed way.

In this regard, the authors should specify which residues are included in the box. They have to report the size of the box, the number of gridpoints along the x,y, and z axis, and the spacing among the gridpoints, the center of the gridbox for allowing the replication of the proposed analysis. The authors can take a look at the docking section of Giangregorio et al., IJBioMac 2022 for having an idea of the necessary parameters to be provided in such a “Method” section, for allowing other researchers to replicate the proposed analysis.

Line 483: what the authors mean with “turned out to be unstable” and how the justify this result.

Line 495: based on which criterion the authors chose “a frame extracted from the first 100 ns”? Please, explain.

CONCLUSIONS

This reviewer would suggest revising the conclusion section in light of the above-reported concerns to be addressed.

Lines 519-521:

See the comments reported above in correspondence of lines 328-330 and discuss in light of the above-cited papers the obtained results.

Line 525:

“ammonium group” should be “trimethylammonium group”?

Round 2

Reviewer 2 Report

All the concerns have been solved in the revised form.

Reviewer 3 Report

The authors addressed all the risen concerns